# Immunogenicity of the 13-Valent Pneumococcal Conjugated Vaccine Followed by the 23-Valent Polysaccharide Vaccine in Chronic Lymphocytic Leukemia

**DOI:** 10.3390/vaccines11071201

**Published:** 2023-07-04

**Authors:** Sabine Haggenburg, Hannah M. Garcia Garrido, Iris M. J. Kant, Hanneke M. Van der Straaten, Fransien De Boer, Sabina Kersting, Djamila Issa, Doreen Te Raa, Hein P. J. Visser, Arnon P. Kater, Abraham Goorhuis, Koen De Heer

**Affiliations:** 1Department of Hematology, Cancer Center Amsterdam, Lymphoma and Myeloma Center Amsterdam, Amsterdam UMC, University of Amsterdam, 1105 AZ Amsterdam, The Netherlandsk.deheer@amsterdamumc.nl (K.D.H.); 2Amsterdam Institute for Infection and Immunity, Amsterdam UMC, University of Amsterdam, 1105 AZ Amsterdam, The Netherlands; 3Department of Infectious Diseases, Center for Tropical Medicine and Travel Medicine, Amsterdam UMC, University of Amsterdam, 1105 AZ Amsterdam, The Netherlandsa.goorhuis@amsterdamumc.nl (A.G.); 4Department of Internal Medicine, St Jansdal Ziekenhuis, 3844 DG Harderwijk, The Netherlands; 5Department of Internal Medicine, Ikazia Ziekenhuis, 3083 AN Rotterdam, The Netherlands; 6Department of Hematology, HagaZiekenhuis, 2545 AA The Hague, The Netherlands; 7Department of Internal Medicine, Jeroen Bosch Ziekenhuis, 5223 GZ ‘s-Hertogenbosch, The Netherlands; d.issa@jbz.nl; 8Department of Internal Medicine, Ziekenhuis Gelderse Vallei, 6716 RP Ede, The Netherlands; raad@zgv.nl; 9Department of Internal Medicine, Noordwest Ziekenhuisgroep, 1815 JD Alkmaar, The Netherlands; 10Department of Internal Medicine, Flevoziekenhuis, 1315 RA Almere, The Netherlands

**Keywords:** chronic lymphocytic leukemia, pneumococcal vaccination, immunogenicity, antibody response

## Abstract

Patients with Chronic Lymphocytic Leukemia (CLL) have a 29- to 36-fold increased risk of invasive pneumococcal disease (IPD) compared to healthy adults. Therefore, most guidelines recommend vaccination with the 13-valent pneumococcal conjugated vaccine (PCV13) followed 2 months later by the 23-valent polysaccharide vaccine (PPSV23). Because both CLL as well as immunosuppressive treatment have been identified as major determinants of immunogenicity, we aimed to assess the vaccination schedule in untreated and treated CLL patients. We quantified pneumococcal IgG concentrations against five serotypes shared across both vaccines, and against four serotypes unique to PPSV23, before and eight weeks after vaccination. In this retrospective cohort study, we included 143 CLL patients, either treated (n = 38) or naive to treatment (n = 105). While antibody concentrations increased significantly after vaccination, the overall serologic response was low (10.5%), defined as a ≥4-fold antibody increase against ≥70% of the measured serotypes, and significantly influenced by treatment status and prior lymphocyte number. The serologic protection rate, defined as an antibody concentration of ≥1.3 µg/mL for ≥70% of serotypes, was 13% in untreated and 3% in treated CLL patients. Future research should focus on vaccine regimens with a higher immunogenic potential, such as multi-dose schedules with higher-valent T cell dependent conjugated vaccines.

## 1. Introduction

Patients with chronic lymphocytic leukemia (CLL) are at increased risk of both invasive pneumococcal disease (IPD) and community acquired pneumonia (CAP) caused by *Streptococcus pneumoniae* [1,2,3]. IPD carries a high case-fatality rate of 15% according to the latest analysis of the European Centre for Disease Prevention and Control (ECDC) [4]. A recent study showed that patients with CLL have a 29- to 36-fold increased risk of invasive pneumococcal disease in comparison with the general population (15/100.000 per year) [3]. Although pneumococcal vaccinations are advised and readily available, vaccination responses are diminished [5,6,7], because CLL and its therapy cause profound immune dysfunction, for example, due to hypogammaglobinemia, CLL-mediated T cell dysfunction, and medication induced immunosuppression [8]. More advanced disease stage, lower baseline IgG antibodies and prior treatment have been shown to negatively influence the humoral response to pneumococcal vaccination in CLL patients [7]. International guidelines therefore recommend vaccination at an early stage of disease, preferably before initiation of treatment [9]. The pneumococcal vaccination schedule recommended by the European Conference on Infections in Leukemia (ECIL) 7 guideline consists of Prevenar13, a 13-valent pneumococcal conjugated vaccine (PCV13), followed 2 months later by Pneumovax23, a 23-valent polysaccharide vaccine (PPSV23) [9,10]. Previous studies have only investigated responses to either PCV13 or PPSV23 alone [5,7,11,12,13,14,15,16,17]. Therefore, knowledge about the immunogenicity of this sequential vaccination schedule in CLL patients is lacking. While responses to PPSV23 alone are poor [13,16,18], immunogenicity of the 7-valent pneumococcal conjugated vaccine (PCV7) and PCV13 are better but still significantly impaired compared with healthy individuals [5,7,14]. Our aim was to assess the immunogenicity of the sequential vaccination schedule of PCV13 followed by PPSV23 in treatment naive and treated CLL patients.

## 2. Materials and Methods

### 2.1. Study Design and Population

As part of routine care, one dose of PCV13 followed two months later by one dose of PPSV23 were administered to adult CLL patients naive to pneumococcal vaccination in one of seven participating hospitals. According to the current local standard of care for immunosuppressed individuals, the response to vaccination was assessed by measuring serotype-specific pneumococcal immunoglobulin G (IgG) antibody concentrations prior to and eight weeks after the complete vaccination schedule (Figure 1). Written informed consent was given by all participants before the start of the study to collect their clinical data and laboratory results for research purposes. Permission from the medical ethics committee was granted.

### 2.2. Data Collection

Clinical data were collected according to a standardized electronic case report form (Castor EDC, Amsterdam, The Netherlands). We retrospectively collected the last measured serum IgG antibody level, lymphocyte number, hemoglobin level and platelet level and demographic parameters. Furthermore, we recorded CLL specific parameters from the electronic patient files: Binet and RAI stage, time since CLL diagnosis, past or ongoing treatment and past intravenous immunoglobulins (IVIG) administration.

### 2.3. Laboratory Methods and Outcomes

Prior to, and eight weeks after vaccination, we quantified serum IgG antibody concentrations against five pneumococcal serotypes shared across both PCV13 and PPSV23 (6B, 9V, 14, 19F, 23F), and against four serotypes unique to PPSV23 (8, 15B, 20, 33F), by a quantitative multiplex immunoassay (Luminex technology), as described previously [19,20,21,22]. This assay identifies serotype-specific anti-capsular polysaccharide IgG antibodies to the pneumococcal serotypes 6B, 8, 9V, 14, 15B, 19F, 20, 23F and 33F. The primary outcome was the serologic response rate (SRR), defined as a ≥4-fold increase among at least 70% of the serotype-specific anti-pneumococcal IgG antibodies, i.e., six or more out of nine serotypes. Secondary outcomes were serologic protection rates (SPR), defined as an antibody concentration of ≥1.3 µg/mL for ≥70% of all measured serotypes, in accordance with the definition by the American Academy of Allergy, Asthma & Immunology (AAAAI) [23]. In addition, since most previous studies used the World Health Organization (WHO) cut-off for protection in infants (≥0.35 µg/mL), we also analyzed our data using this less stringent cut-off to be able to compare our results with previous studies [24]. Other outcomes of interest were factors associated with the SRR and SPR after the completed vaccination schedule.

### 2.4. Statistical Analysis

Univariable and multivariable logistic regression models were used to identify factors associated with the SRR and SPR after the completed vaccination schedule, using the clinical variables of interest: age, sex, time since diagnosis, baseline laboratory measurements (hemoglobin level, total IgG, platelet level, lymphocyte number) and treatment status, and performing a stepwise backward selection method based on *p*-value of <0.1. An α significance level of 0.05 was applied. Means and standard deviations are presented for normally distributed data, and medians and interquartile ranges for non-normally distributed data. Serotype-specific antibody concentrations were non-normally distributed and therefore always presented as medians with interquartile ranges. Wilcoxon signed-rank test was used to compare pre- and post-vaccination antibody concentrations. To compare groups, a Student's *t*-test or a Mann-Whitney U test was performed. Statistical analyses were performed using the IBM SPSS Statistics for Windows, Version 28.0 (IBM Corp, Armonk, NY, USA).

## 3. Results

From February 2021 through September 2022, a total of 143 CLL patients were included. Patient baseline characteristics are summarized in Table 1a. Participants had a mean age of 66 years (SD 9.2), and 34% were female. Median time since CLL diagnosis was three years (IQR 1.0–8.5). Most participants were classified as CLL Binet stage A (72%) and RAI stage 0 (53%). Of the 143 patients, 105 (73.4%) were treatment naive and 38 (26.6%) had previously received or were currently receiving CLL therapy. Of the 38 treated patients, 11 received a single-agent therapy; 14 patients received multiple sequential therapies or combinations of therapies without chemotherapy; 13 patients received chemotherapy and either rituximab, ibrutinib and/or venetoclax (Table 1b). In total, 28 of the included patients had rituximab in their treatment regimen, of whom 9 less than 12 months prior to vaccination. Seven patients received a combination of ibrutinib and venetoclax less than one month prior the first vaccination. Eight patients received IVIG, of whom two before the first vaccination only, four both before and after, and two only after the vaccination series. Prior to vaccination, the median serotype-specific pneumococcal IgG antibody concentrations in the overall cohort ranged between 0.06 µg/mL (serotype 6B and 9V) and 0.26 µg/mL (serotype 14) for the shared serotypes, and between 0.12 µg/mL (serotype 8) and 0.40 µg/mL (serotype 33F) for the PPSV23 unique serotypes (Appendix A). There was a significant difference in pre-vaccination serotype-specific antibody concentrations between treated and untreated CLL patients for serotypes 19F, 23F, 8 and 20 (higher in treatment naive group).

### 3.1. Antibody Concentrations after Complete Vaccination

Serotype-specific antibody concentrations prior to and after the complete vaccination schedule are presented in Figure 2 and Appendix A. On average, a statistically significant increase was observed for all serotype-specific antibody concentrations after vaccination compared with prior antibody levels (*p* < 0.001). However, some patients did not generate measurable antibody concentrations against certain serotypes (Figure 2). Participants without a CLL treatment history obtained significantly higher post-vaccination antibody concentrations compared with treated patients (Table 2; Figure 2).

The relative difference in post-vaccination titers between those two groups was most pronounced for the shared serotypes (Table 2).

### 3.2. Serologic Response and Protection

The overall SRR was 15/143 (10.5%), with better responses in untreated CLL patients (14/105; 13.3%) vs. treated CLL patients (1/38; 2.6%) (Table 3a). The overall SPR increased from 2/143 (1.4%) before vaccination to 15/143 (10.5%) after both vaccinations (Table 3a). None of the patients who received IVIG or rituximab in the 12 months prior to vaccination or during the vaccination schedule obtained a sufficient serologic response. The one treated CLL patient with a serologic response had not recently received chemotherapy (five years prior to the vaccination schedule). Large heterogeneity was observed in the immunogenicity per serotype. Serologic response rates were lowest for serotype 20 (16%) and highest for serotype 9V (40%), whereas the serologic protection rates were highest for serotype 14 and 33F (44% and 40%, respectively), and lowest for serotype 6B (16%) (Table 3b).

We used a stringent cut-off for serologic protection of 1.3 µg/mL, as defined by the AAAAI, as mentioned before. Using the less stringent WHO cut-off for infants (0.35 µg/mL), the overall SPRs were higher, and increased from 12/143 (8.4%) to 40/143 (28%): 4/38 (10.5%) in treated CLL patients versus 36/105 (34.3%) in untreated CLL patients (Table 3a).

### 3.3. Factors Associated with Serologic Response and Protection

In multivariable analysis, treatment status was significantly associated with the SRR, with treated patients having a lower serologic response (OR 0.07 (0.01–0.67), *p* = 0.01). Additionally, a lower lymphocyte count (associated with treatment status) prior to vaccination was associated with a lower SRR (OR 0.96 (0.92–0.99), *p* = 0.02) (Table 4). Other factors, such as age, sex, prior hemoglobin level, platelet count, and total IgG level were not significantly associated with the SRR. In a multivariable analysis with the SPR as the dependent variable, only a lower lymphocyte level prior to vaccinations was a significantly associated factor (OR 0.97 (0.94–1.00), *p* = 0.05). The time-interval between PCV13 and PPSV23 was modestly but significantly associated with both the SRR and SPR in our cohort, OR 1.02 (IQR 1.00–1.05) *p* = 0.04 and OR 1.03 (1.00–1.06), respectively, with longer intervals associated with better responses.

## 4. Discussion

This study among patients with CLL constitutes the largest CLL cohort to date describing the immunogenicity of the internationally recommended sequential PCV13 combined with PPSV23 vaccination schedule. The serologic response and protection induced by this combined schedule is impaired in previously treated CLL patients as well as in treatment naive patients. Nevertheless, CLL patients naive to treatment obtained consistently higher antibody concentrations than treated CLL patients. This is in accordance with previous research, where both vaccines were investigated separately, instead of sequentially [7,14,16,17]. In our study, the observed response to the combined vaccination schedule is much lower than we recently observed in healthy individuals (82%) and other immunocompromised groups (44–58% in patients receiving immunosuppressive drugs for autoimmune diseases), highlighting the severity of immune suppression associated with CLL [21,25]. Similar poor responses were observed after influenza and recombinant hepatitis B vaccination in CLL patients [26,27]. Despite a relatively small number of treated CLL patients in our study, the negative impact of treatment on vaccination response was clear, with better responses in untreated patients (SRR 13.3% vs. 2.6%), which has also been confirmed by others [6]. This emphasizes the importance of vaccinating early in the disease course of CLL. With a 29- to 36-fold higher incidence of IPD in CLL patients compared with healthy individuals, the low vaccination response observed is of concern [3]. No data exist on vaccine effectiveness in CLL patients, i.e., real-world protection against pneumococcal disease after vaccination. Such studies would require a very large sample size and, more importantly, would be unethical given the already existing guidelines that recommend vaccination in this patient group. Based on the poor immunogenicity of the PCV13/PPSV23 vaccination schedule in this study, vaccine effectiveness is likely to be low.

We recently studied the immunogenicity of sequential PCV13/PPSV23 vaccination in various groups of immunocompromised patients (ICP) [21,22,25,28]. In addition to the absence of a PPSV23 booster response after PCV13 vaccination, PPSV23 even seemed to result in a diminished response (hypo response) for several serotypes common to both PCV13 and PPSV23. As we found a modest but positive effect of a longer time interval between the vaccinations, the observed hypo response induced by PPSV23 could potentially be avoided by prolonging the time-interval between the vaccinations.

The T-cell independent PPSV23 directly stimulates B cells and might stimulate pre-existing antigen-specific memory B cells towards terminal differentiation into antibody-secreting cells [29]. It is suggested that this leads to depletion of the pre-existing pneumococcal-specific memory B cell pool., instead of stimulating the formation of pneumococcal-specific immunological memory [29,30], which precludes the potential of a booster effect upon revaccination. However, the currently most frequently circulating pneumococcal serotypes are mainly covered by PPSV23 [3] and to a lesser extent by PCV13, which means that PPSV23 is still important.

Unlike PPSV23, pneumococcal conjugated vaccines, such as PCV13, stimulate the formation of immunological memory, through the addition of a tetanus toxoid carrier protein, which involve CD4+ T-cells in the immune response, leading to the formation of memory B cells. Therefore, potent booster responses after repeated PCV13 vaccination were demonstrated in hematological stem cell transplant recipients [28]. In addition, PCV13 has been demonstrated to have higher clinical effectiveness against pneumococcal disease compared to PPSV23 [31], possibly because high affinity IgG antibodies are formed by the involvement of T-cells in the immune response [32]. Recently, higher valent pneumococcal conjugated vaccines with broader serotype coverage, such as PCV20 and the 24-valent ASP3772, have been introduced [33,34]. The potential of the mentioned booster effect could mean that multiple doses with these higher valent PCVs could result in better serologic protection and ultimately better vaccine effectiveness. This has already been shown in recent COVID-19 research, where both CLL patients naive to treatment, as well as those treated with ibrutinib, were able to produce sufficient antibody concentrations after vaccination with multiple doses of COVID-19 mRNA vaccine, despite very low initial response rates in treated CLL patients after the standard regimen of two doses [35]. In addition, the mentioned 24-valent ASP3772 pneumococcal conjugate candidate vaccine incorporates a multi antigen-presenting system (MAPS) [34]. This MAPS consists of a fusion protein of two highly conserved pneumococcal proteins, which induced enhanced post-immunization responses in ASP3772-vaccinated healthy individuals, via induction of a B- and Th1 cell response, as well as an additional Th17 cell response, which mediates resistance to mucosal colonization of *S. pneumoniae*. Together this might induce development of protective T cell memory [34,36].

A major limitation of this pneumococcal vaccine immunogenicity study and of immunogenicity studies in ICP in general is that the measured immune response does not necessarily translates into real-world protection, also called vaccine effectiveness. However, very large studies are required to investigate this, and such studies are not feasible, nor ethical, to conduct in relatively small patient populations. Thus, studies that prove that a certain vaccine is immunogenic in an ICP group often constitute the best level of evidence that can be acquired.

Although no clear correlate of protection exists in pneumococcal vaccination studies, an antibody concentration of 1.3 µg/mL is currently the most accepted cut-off for protection [23]. We prefer this AAAAI cut-off for protection (1.3 µg/mL), since adults often have baseline antibody levels exceeding the WHO cut-off of 0.35 μg/mL [23,25,26]. Furthermore, it has been suggested that the WHO threshold for serologic protection, might lead to an overestimation of protection against certain serotypes, even in children [37]. Using the cut-off of 1.3 μg/mL, recent studies showed an overall serologic protection of 82% after the sequential PCV/PPSV23 vaccination in healthy adults [21,22]. This is much higher than the 10.5% that we found in this study, which raises the question whether vaccination in CLL patients is at all relevant. In our opinion, the 29- to 36-fold increased risk of IPD in CLL patients [3], probably offsets the 8-fold lower serologic protection rate, favoring vaccination for all CLL patients, regardless of treatment status.

Moreover, serologic protection is only part of the story, since absence of antibody responses might not preclude the induction of a cellular immune response, which was observed in a recent COVID-19 vaccination study showing that patients with hematologic malignancies were able to mount robust T cell responses in the absence of humoral immunity [38]. Although only pneumococcal-specific B cell memory is formed after PCV13 vaccination, a potential role for ASP3772 vaccines in development of pneumococcal-specific T cell memory is suggested [34]. Furthermore, knowledge about the role of NK cells after pneumococcal immunization is lacking. This emphasizes the need for future research into T and NK cell immunity induced by vaccines in CLL patients.

We conclude that, although the sequential PCV13/PPSV23 vaccination is still relevant for CLL patients and should be recommended, we should focus future research efforts on multiple dosed higher valent pneumococcal conjugated vaccines, such as the recently introduced PCV20 and the 24-valent pneumococcal candidate vaccine ASP3772 that has been announced [33,39].

## Figures and Tables

**Figure 1 vaccines-11-01201-f001:**
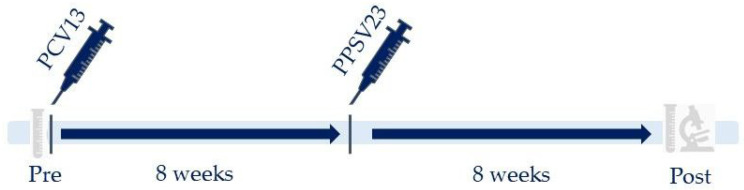
Vaccination and measurement schedule. Patients were retrospectively included when the vaccination and measurement schedule was completed. According to the current local standard of care for immunosuppressed individuals, prior to vaccination, blood was collected to determine serotype-specific pneumococcal antibodies. Post: collection of serum 8 weeks post-immunization; Pre: collection of serum prior to vaccination.

**Figure 2 vaccines-11-01201-f002:**
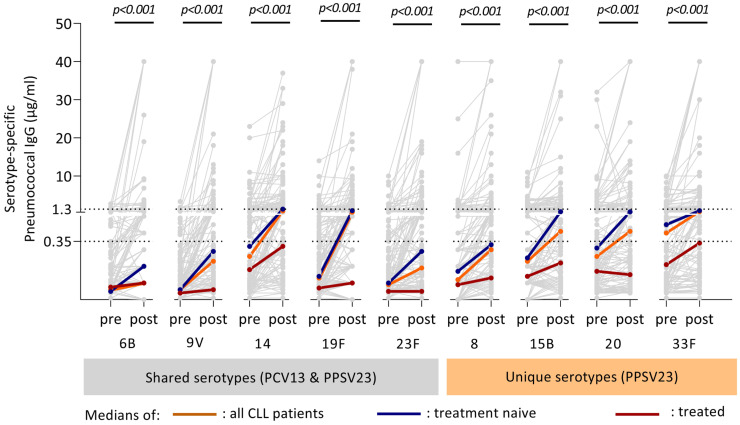
Antibody dynamics of CLL patients after the combined vaccination schedule (PCV13/PPSV23). Antibody concentrations against the five shared serotypes (PCV13/PPSV23) (serotypes highlighted with the grey bar below) and four serotypes unique to PPSV23 (orange bar below serotypes) for each individual CLL patient are shown in the figure. Each grey line indicates the antibody dynamics of one patient from pre- to post-immunization. Median antibody concentration and gradient between pre- and post-vaccination titer are indicated for the overall group (thick orange line), the treatment naive individuals (blue) and treated individuals (red). The upper dotted line indicates 1.3 μg/mL, cut-off for serologic protection, and the lower dotted line indicates 0.35 μg/mL according to the WHO threshold for protection. *p*-values are calculated to determine the median difference between pre- and post-immunization concentration for each serotype, using the Wilcoxon Signed Rank test for related samples. Pre: antibody concentration prior to vaccination; Post: antibody concentration eight weeks post-immunization.

**Table 1 vaccines-11-01201-t001:** Baseline characteristics of included CLL patients. (a) Baseline characteristics. In total, four patients deviated from the protocol and had PPSV23 less than 8 weeks after PCV13 (range 28–49 days after PCV13). (b) Treatment specification for all treated CLL patients. Percentages are calculated for the overall treated population. Combinations of therapies were not necessarily given together in the same treatment regimen. BTK: Bruton’s tyrosine kinase inhibitor; IgG: Immunoglobulin G; IQR: interquartile range; IVIg: intravenous immunoglobulin G; SD: standard deviation.

(a)	
	**overall**
n	143
Age, mean (SD)	66 (9.2)
Sex (female), n (%)	49 (34.3)
**CLL-specific**	
Time since diagnosis (years), median (IQR)	3.1 (1.0–8.5)
Binet stage, n (%)	
Binet A	103 (72.0)
Binet B	32 (22.4)
Binet C	6 (4.2)
Binet unknown	2 (1.4)
RAI stage, n (%)	
RAI 0	76 (53.1)
RAI I	34 (23.4)
RAI II	21 (14.7)
RAI III	7 (4.9)
RAI IV	3 (2.1)
RAI unknown	2 (1.4)
**Treatment status (current or past), n (%)**	
Treatment naive	105 (73.4)
Treated	38 (26.6)
IVIG in last 2 months prior to the study start, n (%)	6 (4.2)
**Laboratory (prior to PCV13)**	
Total IgG (g/L), median (IQR)	6.9 (5.2–9.2)
Hemoglobin level (mmol/L), median (IQR)	8.7 (8.1–9.2)
Lymphocyte no. (×10^9^/L), median (IQR)	18.0 (4.0–53.4)
Thrombocyte no. (×10^9^/L), median (IQR)	211 (147–248)
**Vaccination schedule**	
Days between PCV13 and PPSV23, median (IQR)	59 (56–63)
Days between PPSV23 and final sample date, median (IQR)	41 (34–51)
**(b)**	
**Treatment**	
Treated CLL patient, n	38
Single-agent therapy, n (%)	11 (28.9)
Rituximab ^1^	5
Ibrutinib	4
Venetoclax	1
Chemotherapy	1
Sequential therapies or combination of therapies, n (%)	27 (71.1)
Chemotherapy and …	13
Rituximab	9
Rituximab and ibrutinib	2
Rituximab and venetoclax	1
Ibrutinib and venetoclax	1
Rituximab and …	11
Ibrutinib	4
Venetoclax	4
Ibrutinib and venetoclax	3
Ibrutinib with venetoclax	*3*
Median time since last administration (months), median (IQR) ^2^	
Rituximab	30 (10–47)
Chemotherapy	44 (10–73)

^1^ Monotherapy rituximab was given for auto-immune hemolytic anemia (AIHA) or idiopathic thrombocytopenic purpura (ITP). ^2^ Time since last administration of ibrutinib and venetoclax was not asked.

**Table 2 vaccines-11-01201-t002:** Comparison of pre- and post-immunization antibody concentrations (µg/mL) in CLL patients, stratified by treatment status. Fold change between pre- and post-immunization titers stratified by treatment group are indicated. Differences in pre- and post-immunization concentrations (µg/mL) between groups are calculated using the Mann Whitney U test for unrelated samples.

	Naive to Treatment			Treated			Difference in Pre-Vaccination Concentration	Difference in Post-Vaccination Concentration
	Antibody Concentration (µg/mL)			Antibody Concentration (µg/mL)				
Pneumococcal Serotypes	Prior to Vaccination	After Vaccination	Fold Change (Median)	Prior to Vaccination	After Vaccination	Fold Change (Median)	*p*-Value	*p*-Value
	Median (IQR)	Median (IQR)		Median (IQR)	Median (IQR)			
6B	0.05 (0.04–0.28)	0.16 (0.04–0.87)	2.2	0.08 (0.04–0.19)	0.06 (0.04–0.28)	−0.3	0.951	0.019
9V	0.06 (0.04–0.21)	0.29 (0.06–1.35)	3.8	0.04 (0.04–0.14)	0.06 (0.04–0.20)	0.5	0.320	<0.001
14	0.32 (0.06–1.35)	1.30 (0.23–4.25)	3.1	0.18 (0.04–1.03)	0.32 (0.07–1.40)	0.8	0.281	0.002
19F	0.14 (0.05–0.46)	0.92 (0.17–3.10)	5.6	0.07 (0.04–0.36)	0.10 (0.04–0.57)	0.4	0.048	<0.001
23F	0.10 (0.04–0.46)	0.29 (0.06–2.75)	1.9	0.05 (0.04–0.21)	0.05 (0.04–0.31)	0.0	0.046	<0.001
8	0.17 (0.06–0.56)	0.33 (0.10–1.45)	0.9	0.09 (0.04–0.23)	0.13 (0.05–0.52)	0.4	0.014	0.016
15B	0.25 (0.09–1.04)	0.62 (0.12–2.70)	1.5	0.14 (0.04–0.55)	0.22 (0.04–0.52)	0.6	0.052	0.002
20	0.31 (0.11–1.15)	0.59 (0.18–2.65)	0.9	0.17 (0.05–0.44)	0.15 (0.05–0.77)	−1.2	0.013	<0.001
33F	0.45 (0.11–1.30)	0.93 (0.23–2.90)	1.1	0.21 (0.08–1.10)	0.34 (0.07–1.45)	0.6	0.130	0.016

**Table 3 vaccines-11-01201-t003:** Serologic response and protection rates after the sequential pneumococcal vaccination schedule in CLL patients. (a) Serologic response rate (SRR), defined as a ≥4-fold increase for at least 70% of the serotype-specific anti-pneumococcal IgG antibodies, and serologic protection rate (SPR), defined as an antibody concentration of ≥1.3 µg/mL for at least 70% of all measured serotypes, in accordance with the definition by the AAAAI, stratified by treatment status. As a comparison serologic protection defined by the WHO cut-off of 0.35µg/mL is indicated. (b) Percentages of patients obtaining serologic response and serologic protection in the overall CLL cohort prior to and after the PCV13/PPSV23 vaccination schedule, stratified by serotype.

(a)					
			SRR	SPR	Serologic protection (WHO reference)
		n	n (%)	n (%)	n (%)
All patients		143	15 (10.5)	15 (10.5)	40 (28)
Naive to treatment		105	14 (13.3)	14 (13.3)	36 (34.3)
Treated (ever received therapies)		38	1 (2.6)	1 (2.6)	4 (10.5)
Rituximab		28	0 (0.0)	0 (0.0)	2 (7.1)
Chemotherapy		14	1 (7.1)	1 (7.1)	2 (14.3)
Ibrutinib +/− venetoclax		17	0 (0.0)	0 (0.0)	2 (11.8)
IVIg					
Prior to vaccination		6	0 (0.0)	0 (0.0)	5 (83.3)
During the vaccination schedule		6	0 (0.0)	0 (0.0)	4 (66.7)
**(b)**					
	Serologic response		Serologic protection		
Pneumococcal serotypes	after vaccination		prior to vaccination		after vaccination
	n (%)		n (%)		n (%)
6B	31 (21.7)		5 (3.5)		23 (16.1)
9V	57 (39.9)		5 (3.5)		31 (21.7)
14	45 (31.5)		34 (23.8)		63 (44.1)
19F	50 (35.0)		18 (12.6)		45 (31.5)
23F	40 (28.0)		11 (7.7)		38 (26.6)
8	32 (22.4)		15 (10.5)		33 (23.1)
15B	24 (16.8)		25 (17.5)		41 (28.7)
20	22 (15.4)		27 (18.9)		45 (31.5)
33F	31 (21.7)		32 (22.4)		57 (39.9)

**Table 4 vaccines-11-01201-t004:** Factors associated with the SRR and SPR in sequentially vaccinated CLL patients. Variables significantly associated with the SRR and SPR as dichotomous outcomes determined by univariable and multivariable logistic regression analyses. CI: confidence interval; ns: not significant; OR: odds ratio; ref: reference category in analysis; SPR: serologic protection rate; SRR: serologic response rate.

	Univariable Logistic Regression			Multivariable Logistic Regression		
	OR	95% CI	*p*-Value	OR	95% CI	*p*-Value
**SRR**						
Age (years)	1.03	(0.97–1.09)	0.31			ns
Sex						ns
Male	ref	ref	ref			
Female	0.96	(0.31–2.97)	0.94			
Time since CLL diagnosis (months)	0.99	(0.98–1.00)	0.08			ns
Hemoglobin level (mmol/L)	1.49	(0.74–3.04)	0.27			ns
Thrombocyte no. (×10^9^/L)	1.01	(1.00–1.01)	0.11			ns
Lymphocyte no. (×10^9^/L)	0.98	(0.95–1.00)	0.07	0.96	(0.92–0.99)	**0.02**
Total IgG level (g/L)	1.11	(0.99–1.24)	0.07			ns
Treatment status						
Naive to treatment	ref	ref	ref	ref	ref	ref
Treated	0.16	(0.02–1.33)	0.09	0.07	(0.01–0.67)	**0.01**
**SPR**						
Age (years)	1.01	(0.96–1.07)	0.67			ns
Sex						ns
Male	ref	ref	ref			
Female	0.96	(0.31–2.96)	0.94			
Time since CLL diagnosis (months)	0.99	(0.98–1.00)	0.08			ns
Hemoglobin level (mmol/L)	1.10	(0.56–2.15)	0.79			ns
Thrombocyte no. (×10^9^/L)	1.01	(1.00–1.02)	**0.01**			ns
Lymphocyte no. (×10^9^/L)	0.98	(0.96–1.00)	0.14			ns
Total IgG level (g/L)	1.08	(0.96–1.21)	0.22			ns
Treatment status						
Naive to treatment	ref	ref	ref	ref	ref	ref
Treated	0.16	(0.02–1.33)	0.09	0.07	(0.01–0.67)	**0.01**

## Data Availability

For original data of de-identified individual participant data, please contact Abraham Goorhuis (a.goorhuis@amsterdamumc.nl).

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
