# Peer review of "Immunogenicity of the 13-Valent Pneumococcal Conjugated Vaccine Followed by the 23-Valent Polysaccharide Vaccine in Chronic Lymphocytic Leukemia"

_vaccines, 2023, doi:10.3390/vaccines11071201_

Round 1

Reviewer 1 Report

The article is well structured and presents relevant data for the field. Some small adjustments are needed:

11)      Abstract: Line 23. Please, correct to a “29- to 36-fold increased”. The same in line 235 and 288.

22)      There are scientific names not in italic. Please, review the scientific names.

33)      Materials and Methods: Line 91. “four to eight weeks after vaccination”. It is not clear if the analysis was done four to eight or eight weeks after vaccination, since in the figure 1 is indicated only eight weeks. Please, adjust the figure 1 or the text to make clearer the protocol.

44)      Line 94. Please, provide a reference or explain the multiplex immunoassay protocol.

55)      In Table 1 there are typing and alignment errors. Please check and correct.

66)      Table 2. Shouldn't a point, not with commas, separate the numbers in the fold change?

77)      Table 3. Please, standardize table 3 like the other tables, regarding column colors.

88)      Table 4. Please, correct “Naive tot treatment” in this table.

99)   Line 244. “The T-cell independent PPSV23 directly stimulates B cells, without the formation of immunological memory”. Please, add a reference for this information, since some times a T-cell independent response for polysaccharides can develop immunological memory.

Reviewer 2 Report

The manuscript entitled "Immunogenicity of the 13-valent pneumococcal conjugated vaccine followed by the 23-valent polysaccharide vaccine in chronic lymphocytic leukemia" by Haggenberg et al describes the humoral response to the vaccine regimen to prevent pneumococcal disease in patients with chronic lymphocytic leukemia (CLL). The cohort consisted of CLL patients who were treated and untreated and vaccinated with 13-valent pneumococcal conjugated vaccine (PCV13) followed by the 23-valent polysaccharide vaccine (PPSV23). Blood samples were taken prior to vaccination and 2 months post-vaccination for evaluation of antibodies to the different pneumococcal serotypes. Overall, this study is important because it evaluates the potential vaccine efficacy of the pneumococcal vaccine in a high-risk patient population. The critical findings show the treatment has a major impact on vaccine efficacy with regards to the humoral response. However, there are some issues that authors need to address to enhance the significance of the manuscript.

1) A discussion of the current findings compared to the vaccine efficacy in healthy controls should be included in the discussion. There is likely published data on the effectiveness of the current vaccine regimen. Does CLL impact vaccine responses?

2) Section 3.2 is confusion and difficult to follow. This should be re-written.

3) Table 2 is missing units.

4) How the antibodies were evaluated is not clear. A more detailed method on antibody analysis should be included.

5) Line 135 mentions Table A1, this should be edited. 
